



# Source apportionment of black carbon and combustion-related CO₂ for the determination of source-specific emission factors

Balint Alfoldy[1], Asta Gregorič[1,2], Matic Ivančič[1], Irena Ježek[1], Martin Rigler[1]

[1]Aerosol d.o.o, Ljubljana, SI-1000, Slovenia

[2]Center for Atmospheric Research, University of Nova Gorica, Vipavska 13, Nova Gorica, Sl-5000, Slovenia

*Correspondence to*: Balint Alfoldy (balint.alfoldy@aerosol.eu)

**Abstract.** Black carbon aerosol (BC) typically has two major sources in the urban environment; traffic, and domestic biomass burning which has a significant contribution to urban air pollution during the heating season. Traffic emissions have been widely studied by both laboratory experiments (individual vehicle emission) and real-world measurement campaigns (fleet

emission). However, emission information from biomass burning is limited, especially an insufficiency of experimental results from real-world studies. In this work, the black carbon burden in the urban atmosphere was apportioned to fossil fuel (FF) and biomass burning (BB) related components using the Aethalometer source apportionment model. Applying the BC source apportionment information, the combustion-related $CO_2$ was apportioned by multi-linear regression analysis, supposing that both $CO_2$ components should be correlated with their corresponding BC component. The combination of the Aethalometer

model with the multi-linear regression analysis (AM-MLR) provided the source-specific emission ratios (ER) as the slope of the corresponding BC-$CO_2$ regression. Based on the ER values, the source-specific emission factors (EFs) were determined using the carbon content of the corresponding fuel. The analysis has been carried out on a three-month long BC and $CO_2$ dataset collected at three monitoring locations in Ljubljana, Slovenia, between December 2019 and March 2020. The measured mean site-specific concentration values were in the 3500-4800 ng m⁻³ and 458-472 ppm range for BC and $CO_2$, respectively.

The determined average EFs for BC were 0.39 and 0.16 g/(kg fuel) for traffic and biomass burning, respectively. It was also concluded that the traffic-related BC component dominates the black carbon concentration (55-65% depending on the location), while heating has the major share in the combustion-related $CO_2$ (53-62% depending on the location). The method gave essential information on the source-specific emission factors of BC and $CO_2$, enabling better characterization of urban anthropogenic emissions and the respective measures that may change the anthropogenic emission fingerprint.

## 1 Introduction

Biomass burning (BB) is a significant source of black carbon (BC), brown carbon (BrC) and organic particulate matter, creating a contribution to climate change (Myhre et al., 2013; Tomlin, 2021) and a severe risk to human health (Naeher et al., 2005; Janssen et al., 2011; Sigsgaard et al., 2015; Chen et al., 2017; Brown et al., 2020; Karanasiou et al., 2021). Global hotspots of BB are associated with extensive and persistent wildfires (e.g. deliberate forest burning in Amazonia and Indonesia, accidental



forest and savanna fires in central Africa, North America, the Mediterranean basin and Siberia) (see e.g. Val Martin et al., 2006; Giglio et al., 2013; Smirnov et al., 2015; Chiloane et al, 2017; Healy et al., 2019; Reddington et al., 2019). On the other hand, emission from domestic wood combustion for the purpose of space heating, water boiling or cooking significantly contributes to the BB emission as well, especially in locations of high population density and reduced ventilation (Karagulian et al., 2015; Klimont et al., 2017; Mitchell et al., 2017).

Wood combustion is an important energy source even in well-developed countries, where its emissions add to traffic-related air pollution. The emission characteristics of BB differ from that of internal combustion engines, where the combustion is more complete. Consequently, engines emit less CO, particulate matter and organic compounds per unit of fuel mass, while having higher $NO_X$ emissions compared to BB due to the higher combustion temperature and excess of air (see EEA 2019: 1.A.3.b. versus 1.A.4.a-b.).

Black carbon is a dominant form of particulate matter emitted from fossil fuel (FF) combustion. Diesel engines (before the Euro 5 legislation standard) emit more than 80% of the particle mass (PM) as BC (EEA 2019: 1.A.3.b.). Since diesel vehicles dominate the European vehicle fleet (Cooper, 2020) the high traffic-related BC emission poses significant air quality problems in cities, which is complemented by the BB emission during the heating season.

Due to its harmful health effects, BC emissions of diesel engines are studied intensively worldwide for a long time (see the
original work of Hansen and Rosen, 1990). The BC emission factors have been determined by numerous studies based on laboratory chassis dynamometer tests (Alves et al., 2015; Park et al., 2020), or real-world on-road measurements using either the chasing method (Wang et al., 2012; Ježek et al., 2015; Zavala et al., 2017), or on-board tailpipe measurements by PEMS (Portable Emission Measurement System) (Zheng et al., 2015; Giechaskiel et al., 2019). These tests refer to the emission factors (EF, emitted pollutant per kg of fuel, or km) of individual vehicles, and do not reflect the emission of the entire vehicle
fleet. On the contrary, roadside monitoring offers the opportunity to measure a statistically significant number of vehicles. These measurements are usually carried out in tunnels (Sánches-Ccoyllo, 2005; Ban-Weiss et al., 2009; Brimblecombe et al., 2015; Blanco-Alegre et al., 2020), where elevated pollution concentration levels and negligible interference of other combustion sources (like wood burning) can be assured. In these studies, the EF calculation is usually based on the carbon-balance method (Brimblecombe et al., 2015), when the plume $CO_2$ increment is used to determine the burnt fuel mass.

On the contrary, emissions from biomass burning are not controlled nearly as strictly as from mobile sources. Some studies have investigated specific combustion appliances, providing the emission factors of various pollutants (Querol et al., 2016; Nielsen et al., 2017; Holder et al., 2019; Trubetskaya et al., 2021). The advantages of these studies are the controlled experimental conditions, the information about the combustion parameters (fuel type, combustion temperature, excess of air) and the opportunity to change these parameters, thus EFs concerning a wide spectrum of fuels and combustion conditions were
reported. However, since only a limited number of stoves and combustion scenarios were studied it is difficult to extrapolate these results to a "real-world situation" of a city.

For this reason, other papers focus on the real-world situation and report the atmospheric concentrations of the biomass burning related air pollution. However, since the contribution of the traffic emission always interferes, the pure biomass burning related





air pollution is difficult to study. Consequently, some studies selected specific locations like Glojek et al., (2021) in Loški Potok, Slovenia, that can be considered as a model village of biomass burning emission with negligible contribution of other sources of air pollution. Other studies utilise the integrated source apportionment model of the Aethalometer (Sandradewi et al., 2008) and reported BB- and FF-related BC concentrations separately (see e.g., Dumka et al., 2018; Deng et al., 2020; Liakakou et al., 2020; Mbengue et al., 2020; Milinkovic et al., 2021).

Despite the reliable source apportionment model of the Aethalometer, the adaption of the carbon-balance method from traffic emission studies is problematic due to the lack of the $CO_2$ source apportionment. However, inverse modelling can offer an opportunity to track the air pollution back to their sources. For example, Olivares et al. (2008) applied inverse modelling to retrieve the traffic- and BB-related emission factors of $NO_X$, $PM_{10}$, BC, and particle number.

In this paper we aimed to determine the biomass burning and traffic specific BC emission factors in urban atmosphere during the heating season. We used the carbon-balance method that required the simultaneous source apportionment of BC and $CO_2$ concentrations. The BC source apportionment was performed by the integrated model of the BC monitor (Aethalometer model, AM), while the source apportionment of $CO_2$ was implemented by multi-linear regression analysis (MLR). After the source apportionment of both components, the specific emission ratios (ERs) for BB and FF have been determined and converted to EF values following the carbon-balance method. The measurements were taken during a three-months long monitoring campaign in Ljubljana, Slovenia, during winter 2019-2020. The atmospheric concentration of black carbon was monitored with simultaneous $CO_2$ measurement at three locations of the city with different emission characteristics involving traffic and heating-related emissions, as well as an urban background site.

In the following we introduce our combined Aethalometer model – multi-linear regression analysis (AM-MLR) method that we applied for the determination of the source-specific emission factors. We present the BB- and FF-related emission factors for three different locations of the city. In order to validate the AM-MLR method, an auxiliary measurement campaign was performed during summer, when only fossil fuel combustion was assumed to present. The FF-related emission factors determined during the summer campaign was compared to the result of the AM-MLR method.




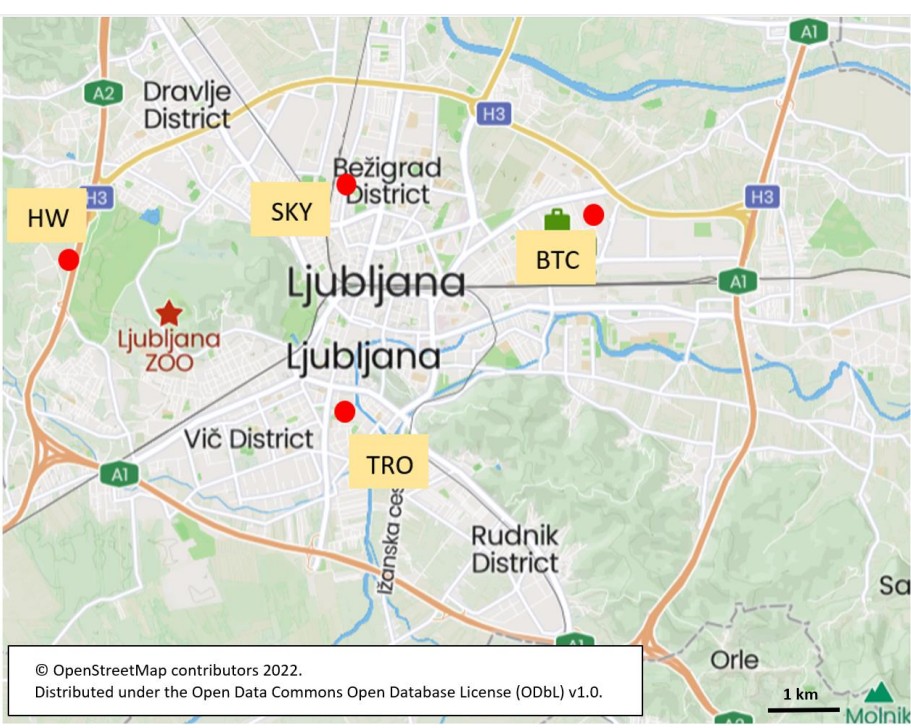

Figure 1: Measurement locations on the map of Ljubljana. HW shows the location of the traffic measurement next to the highway.



## 2 Methods

### 2.1 Measurement sites and instrumentation

The measurement campaign took place from December 2019 to March 2020. Three measurement sites were selected in the city with different microenvironments, source profiles and emission activity (Figure 1). One location was selected in the historical center of the city (Trnovo, TRO), where wood combustion represents the primary energy source for domestic heating during winter. This site is located in the restricted traffic area of the old town, where low direct vehicle emission is expected. The measurement setup was installed in a family house, with the sampling inlet on the roof, 8 m above the ground.

Another location was selected close to major roads and far from biomass burning sources that ensured to measure higher relative contribution of traffic emission. The instruments were installed in a waterproof cabinet in the open recreational area of the Atlantis sport complex in the BTC commercial center (BTC site). The inlet was mounted at 3 m height.

The third measurement location was the atmospheric observatory of Aerosol d.o.o (Skylab, SKY) that is considered to be an urban background location. This location is far from the major roads of the city and not affected directly either by traffic

emission or wood combustion. The sampling inlet was at 10 m above the ground.

The equivalent black carbon (eBC, referred as BC in the following) concentrations were monitored using multi-wavelength Aethalometers (AE33, Magee Scientific/Aerosol d.o.o. Slovenia, Drinovec et al., 2015) that measures the light attenuation of the particle sample collected on a TFE-coated glass filter tape (M8060) at seven wavelengths (370 – 950 nm). The absorption coefficient of the particle sample ($b_{abs}$, $Mm^{-1}$) was obtained by dividing the attenuation coefficient by the multiple scattering

parameter (C=1.39 for M8060 filter tape; Weingartner et al., 2003). The BC mass concentration were generated as the ratio of the absorption and the wavelength dependent Mass Absorption Cross-section parameter ($MAC_\lambda$, $m^2g^{-1}$) provided by the manufacturer. Aerosol size selection was provided at the inlet of the Aethalometer by a cyclone sampling head with $PM_{2.5}$ cut-off diameter. The flow rate was set to 5 l/min and the measurement time resolution to 1 min. The "dual spot" technology enables the real-time loading effect correction, which is especially important when spectral dependence of optical absorption

is used for source apportionment.

The $CO_2$ concentrations were measured by flow-through $CO_2$ sensors (Carbocap GMP 343, Vaisala, Finland). The $CO_2$ sensors were directly connected to the exhaust of the AE33, thus analysing the identical air stream as the Aethalometer. The accuracy of the sensor is 3 ppm below 1000 ppm concentration range, which was the case during the campaign even in the most polluted days. The response time of the sensor is comparable with the AE33, so the 1-minute average signals of BC and $CO_2$ were well

correlated when common sources were measured.

The three measurement systems were compared in the air quality laboratory of Aerosol d.o.o before the campaign. The variation between the AE33 units was below 1% at 1 minute averaging time for both wavelengths used for source apportionment (470 and 950 nm). This precision was expected from the results of Cuesta-Mosquera et al. (2021), who





compared 23 AE33 units and found variation between the measurement results less than 1%. The unit-to-unit variability of the

$CO_2$ sensors the was below 4 ‰ on 1 minute time basis.

## 2.2 Meteorological situation

The measurement campaign started on the 6 of December 2019, during a warming up period that was continued by an unusually warm and dry January and February (Figure 2). Table 1 summarises the basic climatological anomalies comparing to the reference long-term averages of the 1981-2010 period. The average monthly temperatures in Ljubljana during the three-month-

long campaign were warmer than the long-term averages of the 1981–2010 period (2.3 °C, 1.7 °C and 4.8 °C above the long-term average on December, January, and February respectively). February 2020 was the second warmest February in the history of measurements. Usually, January and February are the driest periods of the year, and in 2020, they were even drier than the average. The snow cover was negligible, limited to few days during the measurement period.


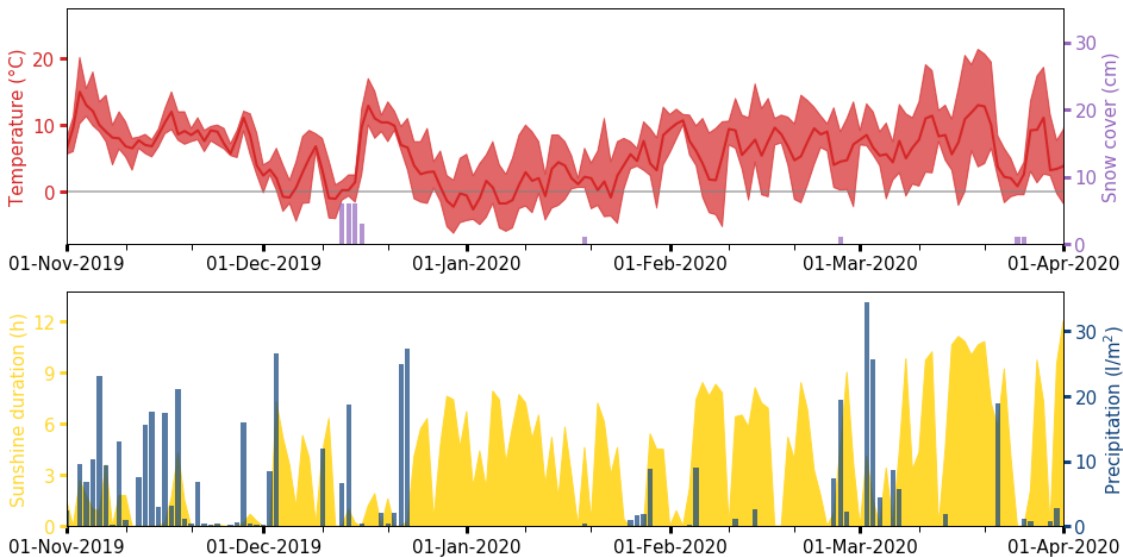

**Figure 2: Time series of minimal, mean, and maximal daily temperatures, snow cover, daily sunshine duration and daily precipitation accumulation in Ljubljana from November 2019 to March 2020.**






**Table 1: Monthly meteorological anomalies relative to reference long-term averages of the 1981–2010 period.**

| Anomaly | 2019 | | 2020 | | |
|---|---|---|---|---|---|
| | November | December | January | February | March |
| Temperature | + 3.1 °C | + 2.3 °C | + 1.7 °C | + 4.8 °C | + 0.7 °C |
| Precipitation | 146% | 121% | 20% | 60% | 119% |
| Sunshine duration | 39% | 156% | 186% | 121% | 119% |



### 2.3 Emission Ratio (ER) and Emission Factor (EF) calculation

Air pollution emission from combustion sources is usually reported with respect to burnt fuel mass and given in fuel consumption-specific emission factor (EF) in g(kg fuel)$^{-1}$ units. Although the combustion is never complete, more than 99% of the fuel carbon content is oxidized to carbon dioxide (EEA, 2019) that can be used as a tracer for fuel consumption estimation. Dividing the pollution concentration by the $CO_2$ increment of the plume, the pollution-to-$CO_2$ emission ratio (ER) can be determined.

The concentration ratio of two particular components of the plume can be calculated by an integrative or derivative way. If the time resolution of the measurement technique of the two components differs significantly, the two concentrations would not be correlated even if they have a common source. In this case, the integrative method is the preferable option for ER calculation. This way the time integral of the concentration peaks are calculated (peak area), and the ratio of the net peak areas (after background removal) provides the ER (see e.g. Ježek et al., 2015). The disadvantage of this method is that the peak identification is arbitrary, and the background definition and removal burden the calculation by an additional uncertainty.

On the other hand, if the time resolution of the two measurement techniques is similar, the recorded pollutant concentrations originated from a common source are correlated in time. In this case a threshold value can be defined for the minimum required $R^2$ of the correlation. Above the threshold $R^2$ the two components are considered to originate from the same source and the slope of regression provides the ER (derivative way). The offset of the regression line depends on the background concentrations that does not need to take into consideration during the calculation.

In our case the BC/$CO_2$ ER was calculated by the derivative method, and later it was transformed to EF using the carbon content of the concerned fuel:

$$EF \left[g(kg\ fuel)^{-1}\right] = ER(\mu g\ m^{-3}/ppm) \cdot \frac{1}{1.82} \cdot \frac{44}{12} \cdot CC, \tag{1}$$

where CC is the carbon content of the fuel that is 0.86 for diesel oil and petrol (Huss et al., 2013), while 0.45 for dry wood (pine tree – Goncalves et al., 2012). The measured $CO_2$ concentration was converted from *ppm* to *mg m*$^{-3}$ using 1.82 *mg m*$^{-3}$*/ppm* conversion factor considering the AMCA atmospheric standard (T=21 $^o$C, P=1 bar) that was also applied by the Aethalometer for the BC concentration calculation. Molecular weight of $CO_2$ (44) and C (12) was used to calculate the carbon mass fraction in $CO_2$.

### 2.4 Source apportionment and source-specific emission ratios

Measurement of the spectrally resolved absorption coefficient provides an insight into the composition of light absorbing particles, allowing to distinguish the highly (and widely) absorbing black carbon (soot) particles from brown carbon (light-absorbing organic aerosols) (Bond & Bergstorm, 2006; Drinovec et al., 2015). Fossil fuel combustion generates mostly pure soot particles that are strong light absorbers over the whole NIR-visible wavelength domain, while particles generated by





biomass burning contain other light absorbing compounds such as brown carbon that have characteristic absorbance bands in the near UV domain (Sandradewi et al., 2008; Helin et al., 2018).

Sandradewi et al. (2008) developed the so called 'Aethalometer model' where the absorptions at 470 and 950 nm wavelengths

were expressed as the sum of the absorptions of the FF- and BB-related BC components ($BC^{FF}$ and $BC^{BB}$), while the ratios of the absorptions at different wavelengths follow a reciprocal power law of the wavelength ratio with a corresponding exponent (called Absorption Ångström Exponent, AAE) of FF- or BB-related BC. In this study, the source-specific AAE pair of 1.15 and 2.1 was used for the FF- and BB-related BC components respectively. The solution of the equation system results in the BB-related absorption at 950-nm wavelength whose ratio to the total absorption provides the ratio of the BB-related BC

concentration.

### 2.4.1 CO₂ source apportionment

In order to apply the carbon-balance method for the source-specific EF calculation, source apportionment of the carbon dioxide is needed as well, which was implemented using the BC source apportionment combined by multi-linear regression analysis (MLR). The method assumes that either the FF- or BB-related $CO_2$ component is correlated with the corresponding BC

component ($BC^{FF}$ or $BC^{BB}$) in the plume. The total measured $CO_2$ can be expressed as follows:

$$CO_2(t) = CO_2^{FF}(t) + CO_2^{BB}(t) + CO_2^{bg},\qquad(2)$$

where $CO_2^{FF}(t)$ and $CO_2^{BB}(t)$ stand for the FF- and BB-related $CO_2$ components of the plume respectively, while $CO_2^{bg}$

represents the background concentration that changes much slower than the combustion-related components; thus, it can be considered constant during a plume event.

Equation (2) can be formulated using the FF- and BB-related BC concentrations and emission ratios ($ER^{FF}$, $ER^{BB}$) as well:

$$CO_2(t) = \frac{BC^{FF}(t)}{ER^{FF}} + \frac{BC^{BB}(t)}{ER^{BB}} + CO_2^{bg},\qquad(3)$$


Or written in an equivalent form:

$$CO_2(t) = \frac{1}{ER^{FF}}\left[BC^{FF}(t) + \frac{ER^{FF}}{ER^{BB}} \cdot BC^{BB}(t)\right] + CO_2^{bg},\qquad(4)$$

Equation (4) expresses that the linear combination of $BC^{FF}(t)$ and $BC^{BB}(t)$ is correlated with the measured $CO_2$, using an appropriate $ER^{FF}/ER^{BB}$ ratio. Our task is to find a particular $ER^{FF}/ER^{BB}$ ratio, which provides the best correlation between the two sides of Eq. (4). After the best correlation was found, the slope of the regression line provides $1/ER^{FF}$, so $ER^{BB}$ can be also calculated. The background $CO_2$ concentration determines the offset of the regression and is not needed to take into





consideration during the calculation. However, the background $CO_2$ provided by MLR is also a valuable information that we
are presenting in this paper.

The MLR analysis is a well-known and widely used method in source apportionment calculations; however, its combination with the Aethalometer model just recently appeared in the literature. Kalogridis et al. (2018) used the source apportionment information provided by the Aethalometer model for the source apportionment of carbon monoxide (CO) in Athens. They compared their result with the linear CO-$NO_X$ model (see there) and concluded that the CO-$NO_X$ model overestimates the BB-
related CO contribution maybe due to the photochemical loss of $NO_X$, while the MLR analysis provided more reliable results. The combination of the Aethalometer model with multi-linear regression analysis (AM-MLR) presented here thus can be a universal technique for source apportionment of any air pollution component that emitted together with BC (for example $CO_2$, CO, NO, $NO_2$, $SO_2$, PM or VOC).

For the application of the MLR analysis the R-statistical package (R Stats, Austria) was used. The correlations were studied in
a running time window with 1-hour duration. During this time interval the background concentration is supposed to be constant, while the FF and BB sources have characteristic emission peaks.

It should be noted that during the 1-hour time window, several FF- and BB-related sources contribute to the measured plume with different ERs. The emitted BC and $CO_2$ concentrations have been averaged out during the MLR, so the ER received from the actual time window refers to the one hour average emission of the sources. The shorter the time window, the shorter the
averaging period, which results in higher variation and wider distribution of the ER values. However, the choice of the time window does not affect the mode of the distribution (the most frequent ER value).

In the following special conditions, the MLR method provided false results, so they were discarded:

1) If the FF and BB components are well correlated ($R^2>0.8$) the MLR method cannot separate the two components and provided similar ERs for the two components. Typically, this was the case when a transported pollution plume was
measured, within the FF and BB components arrived together to the measurement location resulting in correlated concentration increments. In this case the ERs refer about the average BC emission ratio including all the combustion sources (FF+BB) and must be discarded from the results.

2) Plumes dominated by one of the components results in good correlation between the BC component and the total $CO_2$ concentration. In this case the $CO_2$ source apportionment fails, and the total $CO_2$ increment is accounted for the
dominant source, consequently the calculation provides an underestimated EF. For this reason, cases when one of the components correlated well with the total $CO_2$ concentration ($R^2>0.8$) were discarded from the analysis.

3) The maximal P-value for significance criteria was set to $10^{-5}$ for both components. Results exceeding this threshold were discarded from the dataset.

## 2.5 Auxiliary measurements

For the validation of the AM-MLR method a well-defined case is needed with exclusively one type of sources (traffic or wood burning). Since this was never the case during winter, we performed additional measurements during summertime next to the





E61 highway ring around Ljubljana, where the plumes were expected to originate from pure FF emission sources only. A portable monitoring unit was used for the measurement including an AE43 Aethalometer (Aerosol d.o.o, Slovenia) and a Vaisala GMP 343 $CO_2$ sensor, as in the winter campaign. The AE43 is a recently released battery-powered portable version

of the AE33 Aethalometer with identical optical chamber, flow system and operation principle. In addition to its portable setup, the AE43 has a developed firmware and software system that offers improved user experiences with the real-time concentration, and pollution-rose plots.

The measurement station was installed on an overpass road above the highway. The overpass makes a connection between two sections of an unpaved road that has negligible traffic (mostly agricultural vehicles), so practically the highway emission

dominates the concentrations. Due to the fast fluctuation of the concentration and the short lifetime of the pollution peaks emitted by individual sources, 1 second measurement time was used.

Since only FF-related sources were measured, the source apportionment and MLR procedures were not needed. The BC and the $CO_2$ concentration increments were well correlated during the peaks and the slope of the regression was considered as the $ER^{FF}$. Due to the rapid fluctuation of the concentrations, the regression was calculated using a 10-second running time window.

**3 Results**

**3.1 Overview of the measurement results and diurnal cycle of the pollution**

The statistical metrics of the hourly measurement averages at the three locations are summarised in Table 2. The $BC^{FF}$ and $BC^{BB}$ fractions are shown separately, as well as the $CO_2$ concentrations, temperature, and relative humidity. The meteorological parameters were measured at the BTC locations only but can be considered as generally valid values for the whole city area.

It is seen that the traffic related $BC^{FF}$ component dominates the BC load at all locations. The mean $BC^{FF}$ concentrations were 2760, 2200, and 2650 ng m$^{-3}$ at BTC, SKY and TRO locations, respectively; while the corresponding $BC^{BB}$ concentrations were 1470, 1360, and 2180 ng m$^{-3}$. The biggest difference between the FF- and BB-related components can be observed at BTC location (65% vs. 35% of total BC), while the smallest was at TRO (55% vs. 45%), indicating a higher influence of wood combustion in the historical centre of the city.

The spatial variation of the BC components shows an interesting pattern. Relative to the SKY location, the traffic-related FF component is higher by 26% at BTC and 20% at TRO. At the same time, the BB-related BC is higher by 8% at BTC; but 60% at TRO, indicating that this (TRO) location is a definite hotspot in terms of wood combustion. On the other hand, the influence of traffic emission from the surrounding busy roads is still significant at the TRO measurement site even though it is located in a restricted traffic area.

The daily variation of pollution can be followed in the composite day concentration plots shown in Figure 3. The FF- and BB-related BC concentrations are presented separately. It is seen that a pronounced FF peak can be found in the morning at 8:00 local time at all locations, representing traffic emissions during the morning rush hours.



In contrary, the BB sources are more active in the afternoon. After 14:00 the $BC^{BB}$ component starts to increase and reaches its daily maximum in the evening. An especially high evening maximum (3500 ng m$^{-3}$) was found at the BB-influenced TRO

location.


**Table 2. Statistical metrics of the measurements at the three monitoring locations. Mean (with the source-specific percental share respecting the total BC), standard deviation (St. Dev.), their ratio (coefficient of variation, CV), the three quartiles (1Q, Median, 3Q), Minimum and Maximum values as well as their difference (Range) were calculated from hourly averages for the FF- and BB-related BC and $CO_2$ concentrations. Statistical values for the temperature (T) and relative humidity (RH) were given as well for the BTC locations only.**

|  | $BC^{FF}$, ng m$^{-3}$ | | | $BC^{BB}$, ng m$^{-3}$ | | | $CO_2$, ppm | | | T, $^{O}$C | RH, % |
|---|---|---|---|---|---|---|---|---|---|---|---|
|  | BTC | SKY | TRO | BTC | SKY | TRO | BTC | SKY | TRO | BTC | BTC |
| Mean | 2760 | 2200 | 2650 | 1470 | 1360 | 2180 | 458 | 464 | 472 | 3.9 | 82.6 |
|  | 65% | 62% | 55% | 35% | 38% | 45% |  |  |  |  |  |
| St. Dev. | 2390 | 1990 | 2430 | 1510 | 1480 | 2350 | 29.4 | 33.3 | 43.9 | 5.2 | 15.1 |
| CV | 0.87 | 0.91 | 0.92 | 1.03 | 1.09 | 1.08 | 0.06 | 0.07 | 0.09 | 1.3 | 0.18 |
| Min | 30 | 40 | 50 | 10 | 20 | 10 | 407 | 411 | 406 | -7.4 | 18.2 |
| 1Q | 1023 | 749 | 903 | 292 | 219 | 397 | 434 | 436 | 434 | -0.4 | 75.7 |
| Median | 2000 | 1540 | 1950 | 920 | 742 | 1280 | 454 | 458 | 462 | 4.0 | 88.7 |
| 3Q | 3920 | 3030 | 3700 | 2230 | 2060 | 3200 | 476 | 485 | 497 | 8.2 | 94.1 |
| Max | 18900 | 16400 | 20700 | 8470 | 7450 | 14100 | 593 | 613 | 678 | 17.6 | 97.6 |
| Range | 18870 | 16360 | 20650 | 8460 | 7430 | 14090 | 186 | 202 | 272 | 25.0 | 79.4 |



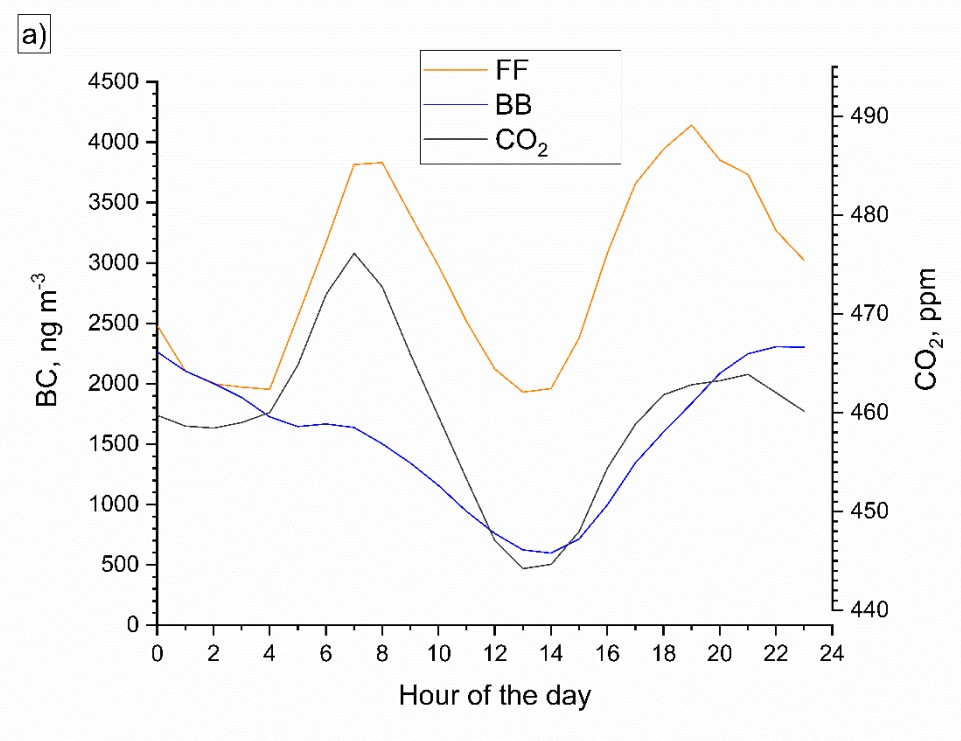

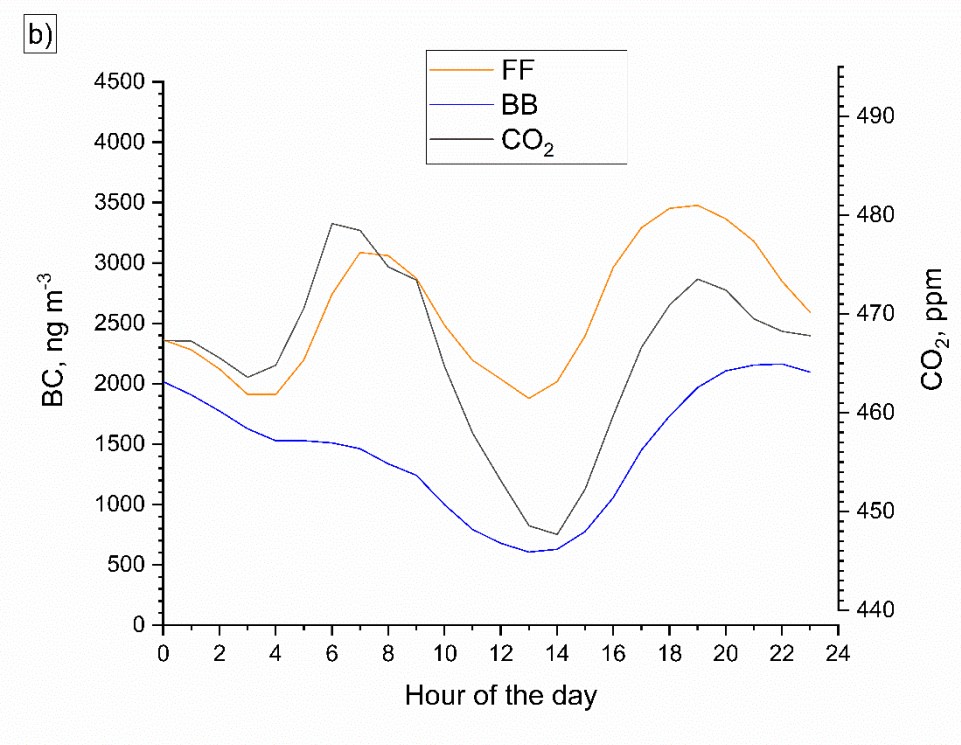



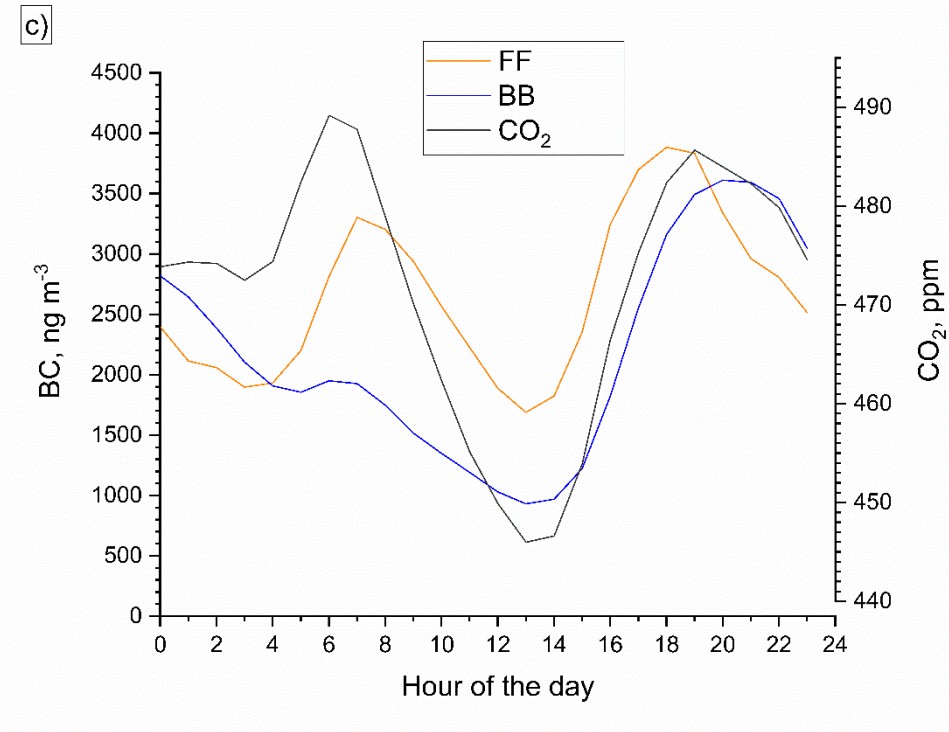


**Figure 3: Diurnal variation of the FF- and BB-related BC components as well as the $CO_2$ concentration at the a) BTC, b) SKY and c) TRO monitoring locations. The time scale represents local time.**





### 3.2 BC/CO₂ emission ratios

Using the BC source apportionment results of the Aethalometer model, the MLR analysis provided the $CO_2$ source apportionment and the source-specific emission ratios. The normalised ER distributions are shown in Figure 4 for the three locations. The distributions are wide and follow a log-normal pattern, ranging from 10 to 1000 ng m$^{-3}$/ppm according to the wide diversity of the sources. Log-normal curves were fitted on the distributions (solid lines in the figures), the parameters of which are summarised in Table 3. The mode and standard deviation that determines a normalised log-normal distribution are

presented in the first two rows of the table. Since the median and mean differ from the mode for a log-normal distribution, these derived parameters are also shown in the last two rows of the table.

The wide distribution of ER can be explained by two main reasons. Firstly, the high variety of sources results in a wide range of emission ratios. For example, the BC emission factor of gasoline vehicles is in the range of 0.001-0.01 g (kg fuel)$^{-1}$, while that of diesel vehicles falls in 0.1-10 g (kg fuel)$^{-1}$ interval (EEA, 2019: 1.A.3.b.). Thus, the measured ER depends on the actual

composition of the traffic, moving towards the higher values during the periods when the contribution of diesel sources (e.g. trucks, buses and goods vehicles) is higher. On the contrary, during periods when the traffic is dominated by personal vehicles, the ER decreases due to the higher contribution of gasoline vehicles.

Regarding the BB sources, the contribution of gas heating to the combustion-related $CO_2$ emission must be taken into account. The BC emission of gas heaters is much smaller than that of wood burning (0.6 g/GJ vs. 74 g/GJ; EEA, 2019: 1.A.4.b), thus

the contribution of gas burning in the $CO_2$ plume dilutes the BB-related emissions. At the same time, the different burning conditions of wood stows from smouldering to high temperature combustion, or the quality of the fuel (wood type, dryness degree) render high divergence of the emission ratios (see low fire – high fire variability in Table 6).

Additionally, the ER distribution widening may be the consequence of a measurement artefact caused by the high $CO_2$ background level. Typically, the combustion-related $CO_2$ increments were measured in the 8-55 ppm interquartile interval,

while the average $CO_2$ background concentration was 437 ppm with 22 ppm interquartile range (see Table 7). This indicates how fast a combustion-related $CO_2$ increment can immerse in the fluctuation of the background during the dispersion of the plume. For this reason, sources with high ER (i.e., low $CO_2$ increment) can be detected close to the sources only, and their relative contribution decreases with increasing distance between the source and the measurement point. Therefore, diluted plumes always provide lower ERs than the direct ones, even if the composition of the sources is similar. Simultaneous

measurement of direct and diluted plumes thus results in wider ER distribution with a lower mode compared to the direct measurement. The same phenomenon leads to lower ER values in well-mixed atmospheres, due to the dispersion of the $CO_2$ emission; while atmospheric inversion favours the detection of low $CO_2$ increments, thus resulting in higher ER.

Both the variation of the sources and the plume dispersion result in a typical diurnal pattern of the ER. Figure 5 shows the diurnal pattern of the (a) FF- and (b) BB-related ER at the BTC location as a typical example.



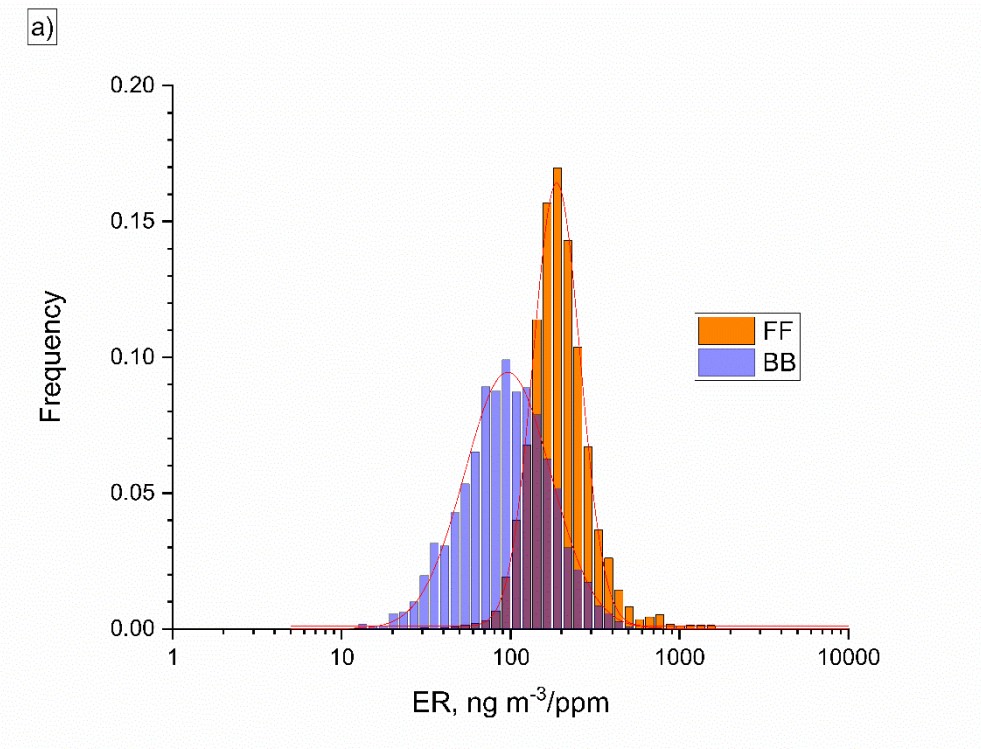

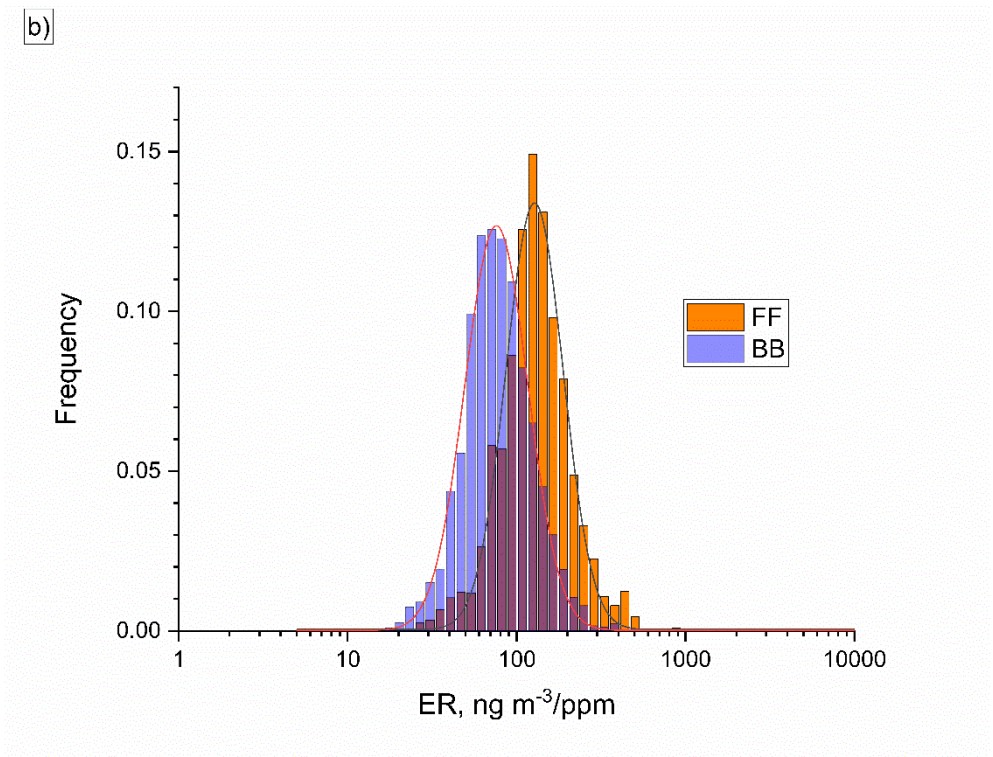

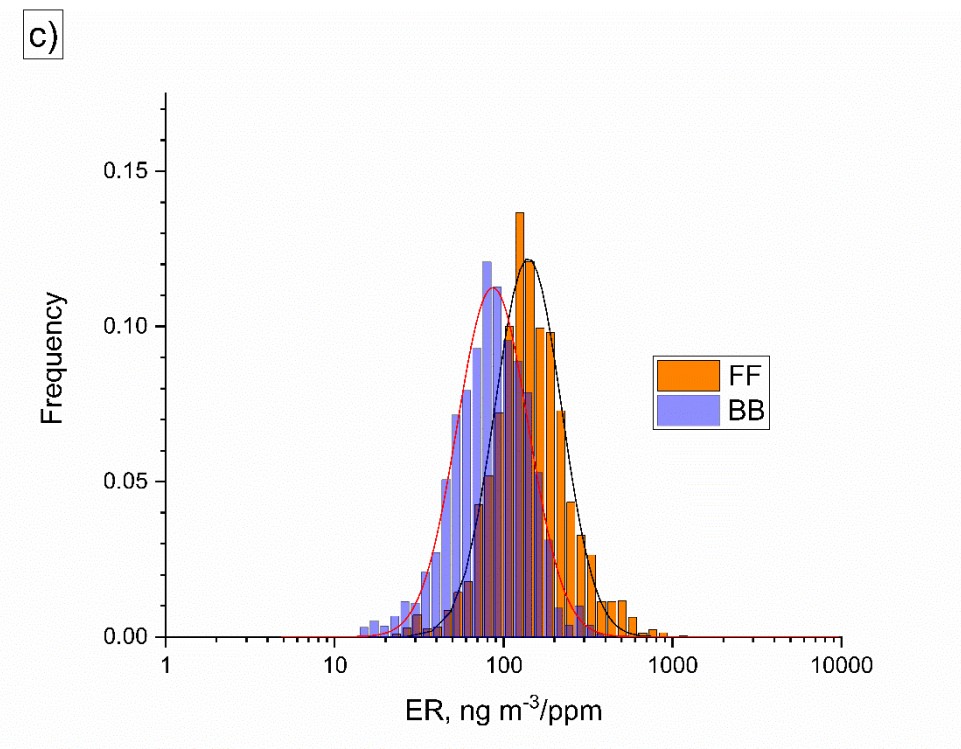

**Figure 4. FF (orange) and BB (blue) related ERs at a) BTC, b) SKY and c) TRO locations respectively. The distributions are**

**normalised to 1. Lognormal distributions (solid lines) were fitted to the results: the parameters are summarised in Table 3.**



**Table 3. Fitting parameters of the lognormal ER distributions (ng m$^{-3}$/ppm) at the three locations of the city. The distributions are normalised to 1. The derived Median and Mean ER values are also shown.**

|          | BTC  |      | SKY  |      | TRO  |      |
|----------|------|------|------|------|------|------|
|          | FF   | BB   | FF   | BB   | FF   | BB   |
| Mode     | 187  | 96.1 | 128  | 75.8 | 140  | 88.5 |
| St. Dev. | 72.0 | 103  | 65.0 | 44.9 | 84.7 | 65.3 |
| Median   | 208  | 136  | 149  | 91.2 | 169  | 112  |
| Mean     | 219  | 161  | 160  | 100  | 185  | 126  |




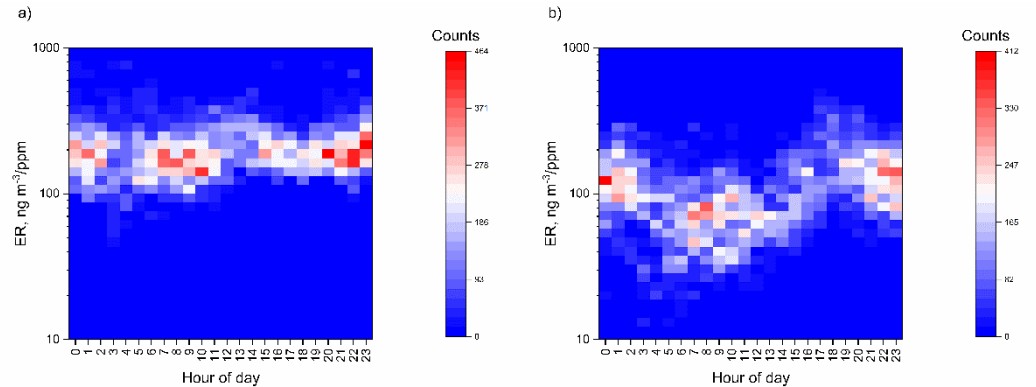

**Figure 5. Diurnal variation of ER$^{FF}$ (a) and ER$^{BB}$ (b) distributions at BTC location (traffic site). Horizontal axes show the hour of the day while the ER is plotted on the vertical axes (logarithmic scale). The pixel colour corresponds to the number of counts in the ER bin.**



It is seen that $ER^{FF}$ varies in a well-defined range between 100 and 300 ng m$^{-3}$/ppm during the day, according to the direct
measurement of the sources (traffic site). On the other hand, $ER^{BB}$ shows a clear diurnal pattern with a minimum during the
morning and maximum in the evening and night. The morning minimum can be attributed to the higher dilution of the pollution
due to atmospheric dynamics or higher relative contribution of gas burning, which shifts the ER distribution down according
to the above discussion. It can also be supposed that the combustion conditions are changing over the night from ignition
through flaming until smouldering phases that leads to different BC release relative to $CO_2$ emission. (For more detailed
relationship between combustion phases and emissions see Shen et al., 2021.)

Similar, but less pronounced diurnal pattern can be observed at the TRO location, while at the SKY location the $ER^{BB}$
distribution does not show significant daily variation (similarly to the $ER^{FF}$ distribution at all the locations).

The wide distribution of the ER values at the BTC and TRO locations can be narrowed by data filtering based on the time.
Figure 6 shows biomass burning ER distributions measured at BTC location between 05:00 and 15:00 (Day) and 16:00-04:00
(Night) separately. Log-normal functions were fitted to the distributions, and the parameters are summarised in Table 4. The
same filtering was performed for the TRO location, while no filtering was applied for SKY location since the ER values did
not show a diurnal variation (parameters from Table 3 have been repeated).

After the time grouping, the daily ER values at the BTC and TRO locations got closer to each other and to the diurnal ER
value of the SKY location (68.7, 76.1, 75.8 ng m$^{-3}$/ppm respectively). Since the SKY location is considered an urban
background location that is not directly affected by either traffic or biomass combustion, we can conclude that diluted and/or
transported plumes were measured at the BTC and TRO locations during the day-time period. On the other hand, significantly
higher ER values were measured during the night-time period (127 and 95.5 ng m$^{-3}$/ppm at BTC and TRO, respectively) when
emissions of the nearby sources dominate the plume composition.





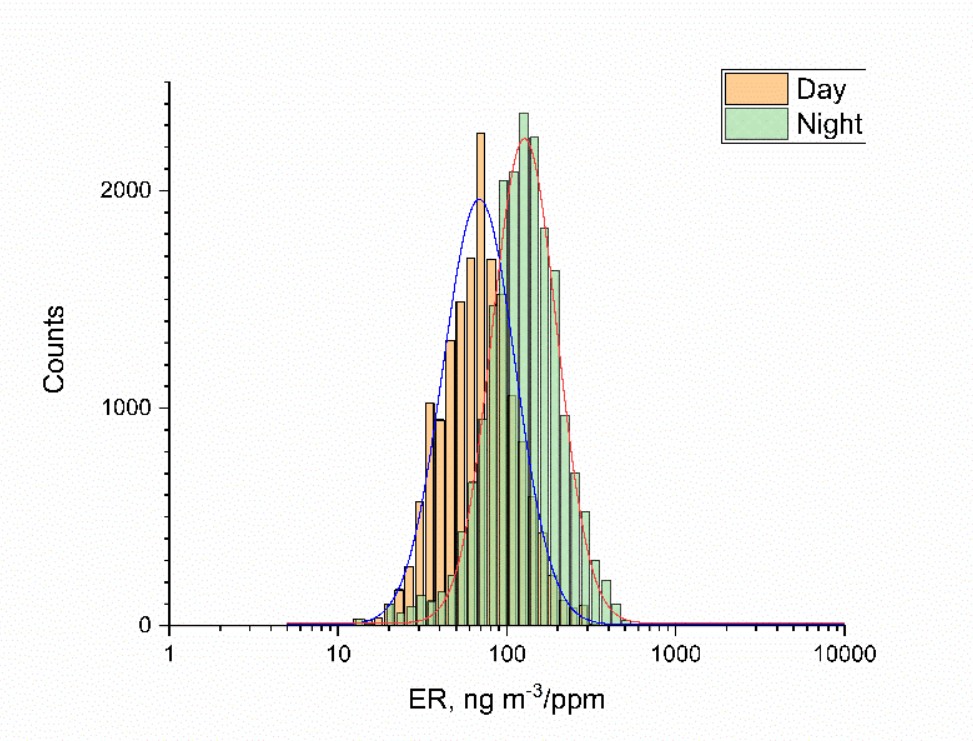


**Figure 6. ER$^{BB}$ distribution at the BTC location during daytime (05:00-15:00) and night-time (16:00-04:00) period. The lines show the fitted lognormal distributions (see parameters in Table 4).**





**Table 4. Fitting parameters of the lognormal ER$^{BB}$ distributions (ng m$^{-3}$/ppm) measured at the BTC and TRO locations during day and night period. The distributions are normalised to 1. The derived Median and Mean are also presented. Values from Table 3 are repeated for the SKY location.**

|  | BTC | | SKY | TRO | |
|---|---|---|---|---|---|
|  | Day | Night |  | Day | Night |
| Mode | 68.7 | 127 | 75.8 | 76.1 | 95.5 |
| St. Dev. | 52.3 | 79.3 | 44.9 | 62.9 | 61.3 |
| Median | 87.8 | 155 | 91.2 | 102 | 117 |
| Mean | 92.3 | 171 | 100 | 116 | 129 |





### 3.3 Emission factors of biomass burning and fossil fuel combustion

The emission factors were calculated from the ERs for biomass burning and traffic using Eq. (1). In Table 6 we show the results from our work together with other results of relevant studies from the literature. Mean values are shown in the table according to the literature data with the interquartile ranges in brackets.

We must note that the comparison of our results with the literature data is still problematic, for two main reasons. First, the artefact caused by the high $CO_2$ background and diluted plume resulted in biased EFs that underestimate the real values. To minimize this bias, we present EFs where the effect of the dilution is most likely to be low (i.e. BTC measurement for FF and BTC & TRO night measurements for BB).

The second problem is that our results represent the average case of numerous urban sources involving low BC emitters (or non-smoking sources) that mostly contribute to the $CO_2$ increment (e.g. gas heating, gasoline vehicles). Thus, our results show lower EFs than of individual sources published in the literature. In the following we discuss our results in the context of the literature data considering the above-mentioned aspects.

### 3.3.1 Traffic emission

The distribution of the traffic-related EF measured at BTC and at the highway is compared in Figure 7. Log-normal fits on the measured data are also shown. It is seen that the EF distribution at the highway site is much wider according to the applied short averaging window (10 seconds) during MLR analysis that allows to detect even individual sources. On the other hand, the two distributions covering each other with similar modus (0.33 and 0.36 g/(kg fuel) at BTC and the highway respectively, see Table 5). This good agreement between the AM-MLR method and the pure FF measurement verifies the validity of the AM-MLR method and indicates that the EF values were not distorted by the dilution effect.

In Figure 7 relevant data from the literature are also shown in scatter plots (see more details in Table 6). Enroth et al. (2016) studied EFs of a mixed fleet in Finland near a highway. Their mean EFs were in the 0.15-0.54 g/(kg fuel) range that overlaps with the EF distribution curve at BTC provided by the AM-MLR method (Fig. 7).

Blanco-Alegre et al. (2020) measured BC EF in a 1 km long urban tunnel in Braga, Portugal. Tunnels ensure well defined conditions for traffic EF measurements with concentrated pollution that mostly originates from vehicle emission. The authors obtained an average EF of 0.31 g/(kg fuel) for the fleet of nearly 56,000 vehicles, whose composition is probably similar to the Slovenian fleet (Cooper, 2020). This value is in a very good agreement with the result of our AM-MLR method at BTC (0.39 g/(kg fuel) average).

Olivares et al. (2008) measured source-specific black carbon concentration in Temuco, Chile by Aethalometer and particle soot absorption photometer (PSAP). They determined the EF for mixed fleet by inverse modelling that gave results of 0.35 g/(kg fuel) mean EF for Aethalometer and 0.61-0.73 g/(kg fuel) EFs for PSAP that fit into our EF distribution.

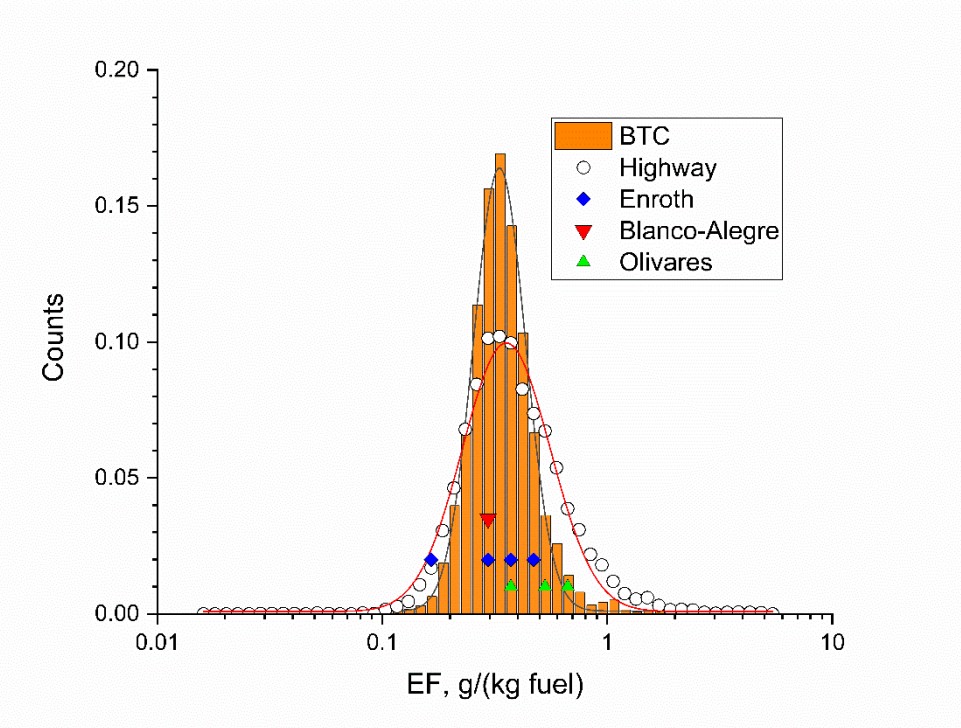

**Figure 7. Distributions of the emission factors originated from traffic at BTC (bar chart) and the highway (scatter plot). Lognormal functions were fitted on the points with parameters summarised in Table 5. Colorised scatter symbols refer the literature data (only X-axes concerned). See more details in Table 6.**




**Table 5. Fitting parameters of the lognormal distributions of the fossil fuel related emission factor (EF, g/(kg fuel)) at BTC and the highway. The distributions are normalised to 1. The derived Median and Mean are also shown.**

|          | BTC  | Highway |
|----------|------|---------|
| Mode     | 0.33 | 0.36    |
| St. Dev. | 0.13 | 0.33    |
| Median   | 0.37 | 0.49    |
| Mean     | 0.39 | 0.56    |






Assuming that BC emissions of gasoline vehicles are negligible comparing to those of diesel engines (as is supported by tailpipe emission measurements – EEA 2019), all the measured $BC^{FF}$ can be attributed to diesel emission. On the other hand,

the diesel emission related carbon dioxide can be estimated based on the share of diesel cars in the vehicle fleet: namely, 36% in Slovenia (National interoperability framework – portal NIO, https://nio.gov.si/nio). This means that the diesel emission related $CO_2$ is roughly 36% of the total $CO_2^{FF}$. The emission factor of diesel engines thus can be calculated by dividing the original EF by 0.36.

In Table 6 the transformed EFs are presented for BTC and highway locations. These numbers refer the diesel EF only and they

are in a good agreement with Brimblecombe et al. (2015), who reported 1.28 g/(kg fuel) diesel EF from a tunnel experiment in Hong Kong. The reported EF values from individual diesel cars (Ježek et al., 2015; Alves et al., 2015; Zavala et al., 2017; EEA, 2019) and individual truck emission monitoring (Ban-Weiss et al., 2009; Dallmann et al., 2011) are in a good agreement with our transformed EF distribution (Figure 8a).

### 3.3.2 Biomass burning

According to the literature data, the biomass burning EF from individual stove emission measurements ranges from 0.063 g/(kg fuel) (Sun et al., 2018) to 0.83 g/(kg fuel) (Holder et al. 2019; Akagi et al. 2011). The wide dispersion of the literature values indicates the high variety of BB EFs according to the stove type and combustion conditions. Figure 8b demonstrates that most of the literature EF data fall above our distribution measured at TRO location. The lower EFs we found here can be the consequences of the contribution of gas combustion sources that are common all around the city. Gas burning emits a very

small mass of aerosol particles compared to wood combustion: but at the same time, it significantly contributes to the $CO_2$ emissions from domestic heating. Since gas combustion for heating probably has the same time pattern as wood combustion (i.e. concentration increments during the evening and cold weather, while drop during midday and warmer periods) the $CO_2$ increments that correlate with the $BC^{BB}$ component partially originated from gas heating. Our method thus cannot uniquely identify EFs from pure wood combustion, but instead refers to the emission factor of the general domestic heating including

non-smoking sources as well. In an ideal case, when the measured sources were exclusively fueled by wood, the heating-related EF would equal with the $EF^{BB}$, otherwise the higher the contribution of gas heating the lower the EF.

However, we also note that the real-world EF data published by Olivares et al. (2008) and the stove emission EF for pine tree by Sun et al. (2018) fall on the low end of the EF distribution measured at the TRO location.






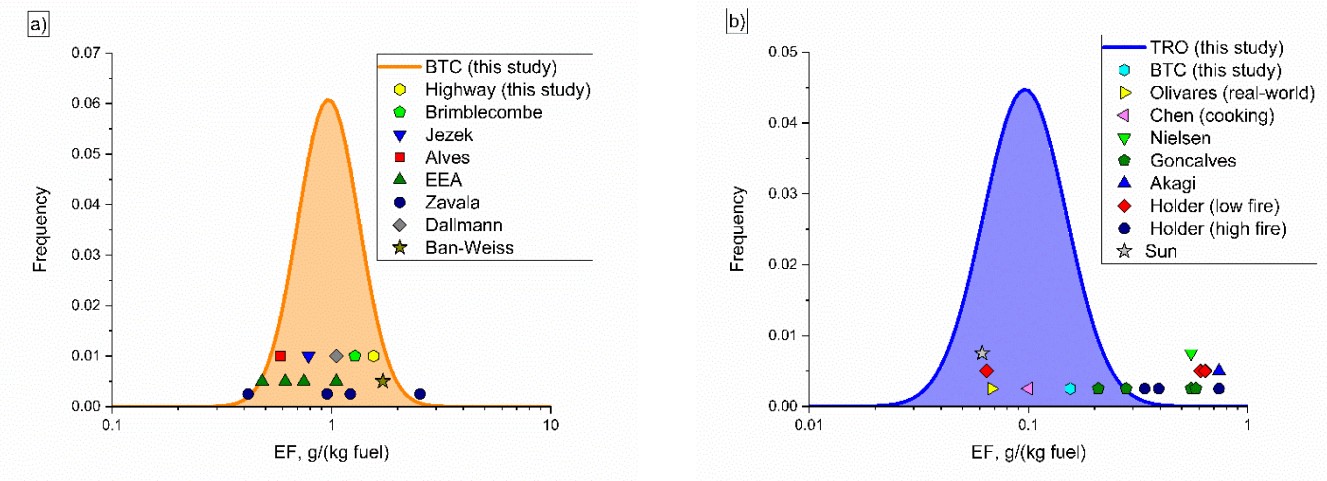

**Figure 8. Distribution of the transformed FF EFs referring the diesel emission at the BTC location (a). Scatter points show the literature data and the highway measurement (only X-axis concerned). b) Distribution of the BB EFs at the TRO location. Scatter points show the literature data and the BTC result (only X-axis concerned). See more details in Table 6.**




**Table 6. Emission factors of fossil fuel and biomass burning sources. Comparison of the results of this study with the literature. Results from the present study are shown as mean values and interquartile range in brackets (q1-q3).**

| Source of data – measurement conditions | Emission factor, g/(kg fuel) | |
| --- | --- | --- |
| | Fossil fuel (traffic) | Biomass burning |
| Enroth et al. (2016) highway study, 4 locations – **mixed fleet** | 0.15; 0.30; 0.43; 0.54 | |
| Blanco-Alegre *et al.* (2020), tunnel study – **mixed fleet** | 0.31 | |
| Brimblecombe et al. (2015), tunnel study – **diesel fleet** | 1.28 | |
| Ban-Weiss et al. (2009), tunnel study – **individual** diesel trucks | 1.7 | |
| Dallman et al. (2011), roadside study – **individual** diesel trucks | 1.07 | |
| Ježek et al. (2015), chasing measurement – **individual** diesel cars | 0.79 (0.36-1.36) | |
| Zavala et al. (2017), chasing meas. – **individual** diesel vehicles | 0.41; 0.94; 1.24; 2.48 | |
| Alves et al., 2015, dynamo chassis study – **individual** Euro4 and Euro3 diesel cars* | 0.59; 0.58 | |
| EEA (2019), dynamo chassis study – **individual** Euro4, Euro3, Euro2, Euro1 diesel cars respectively** | 0.49; 0.62; 0.73; 1.02 | |
| Olivares et al. (2008), street – **mixed fleet**, PSAP | 0.61; 0.73 | 0.074 |
| – **mixed fleet**, Aethalometer | 0.35 | |
| This study, highway – direct EF measurement | **0.56** (0.28-0.59) | |
| Fleet apportionment corrected EF (36% diesel share) | 1.57 (0.79-1.63) | |
| This study, BTC – AM-MLR source apportionment | **0.39** (0.27-0.42) | **0.16** (0.09 -0.17) |
| Fleet apportionment corrected EF (36% diesel share) | 1.08 (0.75-1.16) | |
| This study, TRO – AM-MLR source apportionment | | **0.12** (0.07-0.13) |
| Akagi et al. (2011), open cooking | | 0.83 |
| Chen et al. (2016), cooking | | 0.11 |
| Nielsen et al. (2017), Nordic wood stove (9 kW), birch wood | | 0.62 |
| Sun et al. (2018), pine tree | | 0.063 |
| Goncalves et al. (2012), oak tree, pine tree – fireplace | | 0.30; 0.62 |
| – traditional wood stove | | 0.23; 0.61 |
| Holder et al. (2019), 3 different stoves, spruce wood – low fire | | 0.07; 0.68; 0.72 |
| – high fire | | 0.37; 0.44; 0.83 |

*Converted from mg/km units using $CO_2$ EF from the same study.

**Converted from $PM_{2.5}$ g/km EF using fuel consumption and BC percentage of $PM_{2.5}$ published by the same study.





## 3.4 Source apportionment of $CO_2$ emission

Using the source apportionment of BC and the calculated BC ER values, the BB and FF source-related $CO_2$ components can
be retrieved. By subtracting the total combustion-related $CO_2$ increment from the measured $CO_2$ level, the non-combustion
related $CO_2$ level can be also determined.

Table 7 summarises the statistical metrics of the BB and FF source-related $CO_2$ concentrations as well as the background level
at the three measurement locations. In addition to the absolute mean values of the BB- and FF-related $CO_2$, their relative
contributions to the total combustion-related $CO_2$ concentration are also shown as percentiles.

It is seen that the average background $CO_2$ concentration was the same (~ 436 ppm) at all the locations. On the other hand, the
source apportionment of the combustion-related $CO_2$ shows significant variation according to the environmental conditions of
the locations. At the BTC location the FF-related $CO_2$ component is slightly lower than the BB component (47 vs. 53%), while
at the TRO location, the BB emission dominates the $CO_2$ level (62%).






**Table 7. Source apportionment of the combustion-related $CO_2$ and the background level (Bg) at the three monitoring locations, as well as at the highway (FF component only). The Mean (with the percental share respecting the total combustion-related $CO_2$), standard deviation (St. Dev.), their ratio (coefficient of variation, CV), the three quartiles (1Q, Median, 3Q), Minimum and Maximum values as well as their difference (Range) were calculated from hourly averages for the FF- and BB-related $CO_2$**

**concentration increments.**

| $CO_2$, ppm | BTC | | | SKY | | | TRO | | | HW | |
|---|---|---|---|---|---|---|---|---|---|---|---|
| | FF | BB | Bg | FF | BB | Bg | FF | BB | Bg | FF | Bg |
| Mean | 19.9 | 22.4 | 437 | 21.8 | 27.7 | 435 | 25.9 | 41.9 | 437 | 34.5 | 498 |
| | 47% | 53% | | 44% | 56% | | 38% | 62% | | 100% | |
| St1. Dev. | 16.1 | 16.4 | 11.2 | 17.2 | 18.7 | 9.67 | 20.8 | 28.5 | 16.9 | 18.1 | 22.6 |
| CV | 0.80 | 0.73 | 0.02 | 0.79 | 0.68 | 0.02 | 0.80 | 0.68 | 0.04 | 0.52 | 0.04 |
| Min | 2.67 | 3.54 | 422 | 2.30 | 6.2 | 424 | 2.68 | 8.19 | 418 | 2.67 | 440 |
| 1Q | 8.29 | 9.89 | 428 | 9.63 | 13.9 | 428 | 11.9 | 20.6 | 424 | 21.7 | 482 |
| Median | 15.7 | 18.7 | 434 | 17.1 | 22.8 | 433 | 20.2 | 35.2 | 433 | 31.5 | 493 |
| 3Q | 26.4 | 30.2 | 443 | 29.0 | 36.2 | 439 | 33.3 | 55.7 | 446 | 43.5 | 512 |
| Max | 148 | 171 | 492 | 130 | 159 | 496 | 143 | 198 | 511 | 286 | 587 |
| Range | 146 | 167 | 70.0 | 128 | 153 | 72.0 | 140 | 190 | 94.2 | 284 | 147 |





## 4 Conclusions

Atmospheric concentrations of black carbon and $CO_2$ were monitored real-time at three urban locations in Ljubljana, Slovenia

that had different impacts of traffic and wood-burning during the winter heating season. The source-specific BC concentrations from the Aethalometer model were used to apportion the combustion-related $CO_2$ by coupling a multi-linear regression method. The analysis presumed two combustion-related sources, namely domestic heating (biomass burning) and traffic (fossil fuel combustion). The combined AM-MLR method provided consistent and realistic "real-world" emission ratios and emission factors for the three measurement locations. The method can be further generalised for source apportionment of other

combustion-related components that of EFs can be later determined. Information about the source specific EFs helps to estimate the pollution emission based on the fuel consumption.

The specific conclusions are the follows:

1. The traffic-related $BC^{FF}$ concentration was higher than $BC^{BB}$ at all locations. The smallest difference was found at TRO (wood combustion site), while the largest difference was obtained at BTC (traffic site). In contrast, the heating

related $CO_2$ concentration were higher at all locations.

2. The determined ERs follow a wide log-normal distribution according to the variety of the fuels (from the non-smoking gasoline or natural gas to the BC producing diesel oil and wood) and sources (DPF equipped vs. conventional diesel vehicles; different types and conditions of wood stoves), as well as combustion conditions (high temperature, excess of air vs. low temperature, deficit air conditions). Also, it was shown that the distances of the sources affect the ER,

since the relative contribution of high ER sources (means low relative $CO_2$ emission) are lower for higher distances due to the dilution of the related $CO_2$ increment.

3. Using the literature data of the carbon content of the fuels (diesel oil vs. wood) the related emission factors (EF) were determined. The determined mean traffic-related EF (0.39 g/(kg fuel)) is in a good agreement with published EF values for a mixed traffic fleet. Using the relative ratio of gasoline and diesel fleet for Slovenia, the diesel emission

related EF could be calculated (1.08 g/(kg fuel)) that is in good agreement with diesel emission factors published in the literature.

4. The BB-related mean EF (0.16 g/(kg fuel)) is lower than the majority of the relevant literature data values reported for individual stoves. This is due to the $CO_2$ contribution of other, non-smoking combustion sources (i.e. gas heating).

5. The AM-MLR method was validated by direct traffic emission monitoring next to the highway during summertime,

when only traffic-related sources were most likely sampled. Thus, the FF-related emission factors could be directly determined without source apportionment. The similarity of the modes of the two distributions indicates that the AM-MLR method provided reliable results.



**Data availability**

Data presented in the paper are available at the authors upon request.

**Authors contribution**

AG and IJ designed the experiments with the supervision and guidance of MR; while BA, MI and AG carried out the measurements. BA developed the methodology for $CO_2$ source apportionment and EF calculations. AG performed the MLR analysis by the R-statistical software package. BA prepared the manuscript with contributions from all co-authors.

**Competing interests**

At the time of the research, the authors were employed by the manufacturer of the Aethalometer instruments, used to measure black carbon concentration in the study. The funding sponsors had no role in the design of the study; in the collection, analyses, or interpretation of data; in the writing of the manuscript; or in the decision to publish the results.

**Acknowledgements**

This research was funded by the Ministry of Economic Development and Technology of Slovenia (grant number: C2130-19-096947) and the Slovenian Research Agency (J1-1716 D). Antony Hansen is thanked for his comments and corrections that helped to finalise the manuscript.

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
