# Peer review of "Source apportionment of black carbon and combustion-related CO2 for the determination of source-specific emission factors"

_Atmospheric Measurement Techniques, 2022_

## Referee Comment (RC1)

The study includes an investigation about the source apportionment of black carbon and combustion-related CO2 in order to determine source-specific emission factors. The analysis was carried out on a three-month long BC and CO2 dataset collected at three monitoring locations in Ljubljana. The authors analysed BC and CO2 concentrations. I highly recommend publishing the revised manuscript. I list below some comments and additions that would substantially improve the manuscript.

**Line 172:** Is all that burns pine tree? If not, I suggest including variability in the CC for other species.

**Line 172:** "the source-specific AAE pair of 1.15 and 2.1 was used for the FF- and BB-related BC components respectively" Why these values? Justify them.

**Line 232:** Do you think the model is applicable to other cases? I think there are a lot of discards, for example a city with gasoil and biomass heating. I like the approach of the study very much, but I think it is applicable to few cases, so I think it should be commented on.

**Line 277:** Add refs with same pattern, e.g.
https://www.sciencedirect.com/science/article/pii/S0169809521005366

**Line 232:** Do you think the model is applicable to other cases? I mean, I think there are a lot of discards, for example a city with gasoil and biomass heating. I like the approach of the study very much, but I think it is applicable to few cases, so I think it should be commented on.

**Figure 3:** Change the $CO_2$ dashed lines to make the figures easier to read.

**Line 345:** ABL data available? I believe that obtaining the ventilation coefficient (indicated in the previous ref) would provide information about pollutant dispersion. A model to obtain ABL is available at: https://www.ready.noaa.gov/READYamet.php

---

## Author Response (AR1)

The authors thank for the anonymous referees their valuable comments and ideas. We did our best to answer the questions and improve the quality of the manuscript. Please find our responses below.

**Referee #1**

**R: Line 172:** Is all that burns pine tree? If not, I suggest including variability in the CC for other species.

**A:** We added a sentence and a reference about the variability of CC of wood (lines 191-194 in the revised manuscript). Also, we justified why we used 0.45 that value is representing the most frequently used pine and oak trees.

**R: Line 172:** "the source-specific AAE pair of 1.15 and 2.1 was used for the FF- and BB-related BC components respectively" Why these values? Justify them.

**A:** The default AAE values set by the manufacturer are 1 and 2 for FF and BB, respectively. However, as presented in the literature, specific source related AAE values can vary with measurement location, since they reflect the characteristics of vehicle fleet and wood burning source and burning conditions (Zotter et al., 2017). We revised these default parameters based on our dataset. Regarding the FF component, we measured AAE of 1.15 during the summer campaign where the contribution of biomass burning was negligible. The BB-related AAE was chosen as the maximal AAE measured at TRO location during night, when the BB related BC was maximal, and contribution of fossil fuel combustion is supposed to be minimal. We added an explanation to the relevant section (line 210).

**R: Line 232:** Do you think the model is applicable to other cases? I think there are a lot of discards, for example a city with gasoil and biomass heating. I like the approach of the study very much, but I think it is applicable to few cases, so I think it should be commented on.

**A:** We agree with the Reviewer. The high number of oil stove emission would make the model fail since we cannot distinguish between the FF component used as car fuel or heating purposes. We add a sentence that describes the limitation of the model (see lines 239-241). The interference of gas heating was considered and discussed in lines 357-359, and lines 474-482.

**R: Line 277:** Add refs with same pattern, e.g. https://www.sciencedirect.com/science/article/pii/S0169809521005366

**A:** The Authors thank for the recommendation; the relevant reference has been added (line 243).

**R: Figure 3:** Change the CO2 dashed lines to make the figures easier to read.

**A:** Done.

**R: Line 345:** ABL data available? I believe that obtaining the ventilation coefficient (indicated in the previous ref) would provide information about pollutant dispersion. A model to obtain ABL is available at: https://www.ready.noaa.gov/READYamet.php

**A:** The Authors thank the idea. The ABL data and the wind speed have been taken from the NOAA database and the corresponding ventilation coefficient (VC) has been taken into consideration. A new figure has been added to the manuscript (Figure 3), where the distribution of VC data and the corresponding BC concentrations are presented. Based on Figure 3 two sub-dataset can be defined, 1) when VC < 3200 $m^2s^{-1}$ considered as low ventilation, and 2) VC > 4600 $m^2s^{-1}$ considered as well-ventilated case, when the atmospheric dynamics (advection and vertical convection) dilute the concentration of the pollution. The EF values were separated according to these sub-cases. Figure 6 has been replaced by EF distribution plots regarding the two ventilation cases for both components and all locations. In the same time Figure 5 in the old version has been discarded since it does not represent new information.

**Ref #2**

**R: Line 69:** Correct 'adaption' with 'adaptation'.

**A:** Done

**R: Lines 104, 107, and 110:** The sampling inlet altitudes above the ground (8 m, 3 m, and 10 m) are different in different locations with known background conditions. Is there any basis for choosing the different sampling inlet altitudes above the ground? How did the authors estimate the optimal altitude of sampling inlet in different locations?

**A:** We chose the sampling height suitable for the build environment of the location. We ensured that the inlet was above the height of the surrounding buildings thus the access of air masses from all directions was possible. More explanation was added to the section (lines 107-115).

**R: Line 115:** Is the multiple scattering parameter (C=1.39) dependent on the type of filter tape alone or do any other factors also influence the same?

**A:** The C value depends on the filter tape and on the cross-sensitivity to scattering. Several studies were published about the multiple scattering coefficient C in the last two years, addressing different filter tapes and different Aethalometer models. Recently, the wavelength and site dependence of the multiple scattering coefficient C was studied for the M8060 filter tape (used in the AE33) in a study by Yus-Díez et al. (2021). They reported the wavelength dependence of C value was not significant in urban environments. Since we don't report absorption coefficients in this study, we have considered the manufacturer's default C value of 1.39 and corresponding MAC value. The above reference has been added to the paper.

**R: Line 116:** Will the wavelength-dependent Mass Absorption Cross-section parameter (MAC) provided by the manufacturer be applicable in all background conditions of sampling? Did the authors attempt to determine the wavelength-dependent MAC values to check the consistency with the values given by the manufacturer?

**A:** The MAC value was recently validated in Ljubljana and the authors found good agreement with the manufacturer's value (Ogrizek et al., 2022). This sentence with the reference of the mentioned paper have been added to the relevant section (line 123).

**R: Line 118:** Can we use the same flow rate and time resolution for the three chosen locations with varied background sources of influence and for different altitudes of sampling inlets?

**A:** All locations were affected by the same urban background air pollution. The three locations differ only in the composition of the direct sources (traffic, heating, or no direct sources). Thus, the measured concentrations (BC and CO2) were similar at the three locations, so same flow rate and averaging time were required.

**R: Line 119:** Provide reference to " ... real-time loading effect correction ...".

**A:** Done.

**R: Line 156:** Can you please illustrate the integrative or derivative way of calculating the concentration ratio of two particular components of the plume?

**A:** The referred section introduced the two different ways of concentration ratio calculation. The derivative way is described in the section in details. The integrative way is introduced in the referred literature (Jezek et al., 2015) for one source (fossil fuel). However, we rewrote the misunderstandable sentence (see line 173) and added more details about the method (lines 184-186).

**R: Line 164:** What is the threshold R2 correlation value considered in this study? What is the basis for choosing this value?

**A:** We set 0.9 as a threshold value for the correlation coefficient. Clarification and more details have been added between lines 256-260.

**R: Line 174:** What is AMAC? Please provide reference to the conversion factor for converting CO2 concentration from ppm to mg m-3.

**A:** AMCA stands for Air Movement and Control Association International Inc. that introduced an atmospheric standard of T=21.11 ºC and P=1013.25 mbar. Both, the ppm → mg m$^{-3}$ conversion and BC concentration was calculated according to this standard. The conversion from ppm to mg m$^{-3}$ generally done by the following formula:

C(mg m$^{-3}$)=0.041354*C(ppm)*molecular weight (see e.g. https://cfpub.epa.gov/ncer_abstracts/index.cfm/fuseaction/display.files/fileid/14285). The factor of 0.041354 corresponds to the chosen standard (here AMCA: T=21.11 ºC, P=1013.25 mbar). We don't think that a reference is needed for this conversion since it is well known and widely applied in atmospheric science. However, the abbreviation of AMCA has been explained in line 193.

**R: Line 193:** What do you mean by multi-linear regression analysis (MLR)? Provide a reference to the method.

**A:** The general description of the MLR method can be found in mathematical or statistical books. Here we introduced a simplified version of the method involving two components only. The method has been described in the beginning of Section 2.4.1 (lines 215-238). We believe that the reader can repeat the same calculations applying Eq. (2)-Eq. (4) or using different statistical software, with built-in method.

**R: Line 201:** Do you apply the MLR method during a plume event? Until this line, there is no mention of how the plume event is detected or identified.

**A:** Plume events were not predefined, they were identified by the MLR method. The BC-CO2 correlation was bad out of a plume event, and it exceeded the threshold $R^2$ in the case of well-defined coincident peaks only. Clarification sentences have been added between lines 184-186 and lines 257-258.

**R: Equations 3 & 4:** For which wavelength do the BCFF and BCBB correspond? Provide clarity.

**A:** The 880 nm wavelength was used for BC concentrations according to the manufacturer's recommendation, while 470 and 950 nm wavelengths were used for the source apportionment. We added the information into lines 134-135.

**R: Equation 4:** What are the units of BCFF, BCBB, CO2bg, and CO2? Do they have the same units?

**A:** In this paper we reported BC in ng m-3 units, while CO2 in ppm units (see Table 2). Consequently, the units of the ER values are ug m-3/ppm (see Eq 1).

**R: Line 210:** What are the upper and lower bounds of the ERFF/ERBB ratio?

**A:** In terms of average values the $ER_{FF}/ER_{BB}$ ratios are about 1.4-1.6 depending on the location (Table 3). On the other hands the $ER_{FF}/ER_{BB}$ ratios calculated from the 60 mins running window fall into a wide interval from 0.36 to 18.4. However, we don't think that the distribution of $ER_{FF}/ER_{BB}$ ratios is important to present in the manuscript.

**R: Figure 4:** If the wide distribution of Emission Ratios (ERs) is because of varied number of sources and the actual composition of the traffic,

**A:** We don't fully understand the question. Detailed description of the wide ER distribution can be found from line 348 to line 369.

**R: Line 349:** What do you mean by '... different BC release relative to CO2 emission"?

**A:** The sentence has been revised (see lines 356-358).

**R: Figure 5:** It would be nice if the diurnal variations of ERFF and ERBB are included for the other two locations (TRO and SKY) too.

**A:** According to the recommendation of the other Reviewer we replaced the day-night variation plot (Figure 6) by a low-high ventilation plot that describes better the atmospheric effect on the ER. The new Figure 6 shows the EF distributions for both ventilation cases at the three locations of the city. Besides this the old Figure 5 does not provide new information, so we discarded it.

**R: Line 353:** It was mentioned that ER values at BTC and TRO locations can be narrowed by data filtering based on the time. Can you please elucidate what data filtering you rereferring to? Please provide clarity. On the subsequent line 356, you were referring that the same filtering was performed for the TRO location while no filtering was applied for SKY location since the ER values did not show a diurnal variation. This means that if the diurnal variation is not seen, then the data filtering is not performed. Can you please provide the physical basis and accuracy for following this type of approach?

**A:** This comment is related to the previous one (Figure 5). The data filtering (or grouping) that is presented in the revised manuscript was applied based on the ventilation category of the data point (low or high ventilation case). We performed the data grouping for all locations and showed that there was no shift in the EF distribution at SKY location (see Figure 6b). We revised the whole section and gave a detailed explanation of the data filtering/grouping method. See the sections from line 385-405, Figure 6 a), b), c) and Table 4.

**R: Line 380:** The usage of 'artefact' in the statement is unwarranted/unclear.

**A:** The referred section (line 405-410 in the revised manuscript) has been rephrased.

**R: Line 385:** "Lower EFs than those corresponding to individual sources published in the literature". How accurate are your EFs estimated with the AM-MLR method? It would be nice if the authors have attempted to test the AM-MLR method first with the individual defined sources (as per the literature) to ascertain the consistency of estimated ERs with those already reported. This will ensure the accuracy of ERs estimated with the AM-MLR method.

**A:** The Reviewer is right, we have to validate our method with single, well-defined sources (only FF emission for example), and this is what we did in the course of the summer highway experiment. We found that the fossil fuel related average EF value is in a good agreement with that of we obtained by the AM-MLR method (see lines 429-434). Unfortunately, we could not carry out a similar experiment for wood burning, since the presence of fossil fuel sources could not be avoided in the city.
Comparing our results with emissions of individual sources is problematic since we were measuring a mixed case (real-word conditions). On the other hand, the distribution of our EF values overlaps with the literature data of individual sources that have also high variability. We believe that the presented agreement with the literature data supports the reliability of our method. Moreover, some of the literature data are in a good agreement with the main values of our calculation. The limitations during the comparison with the literature data is described at lines 405-410, lines 456-466, and lines 471-479.

**R: Line 391:** Correct 'modus' with 'modes'.

**A:** Done.

**R: Lines 439-440:** It was stated that AM-MLR method cannot uniquely identify EFs from pure wood combustion (ideal case) but instead refers to the EFs of the general domestic heating including non-smoking sources as well.

**A:** Oppositely, we stated that our AM-MLR method can identify pure wood combustion related EF only if gas heating is not affecting the CO2 concentration (ideal case). Otherwise, gas heating interferes the method, because we cannot distinguish between CO2 coming from wood heating or gas heating. The gas heating related CO2 contribution results in lower BB EF. Further $^{14}$C isotopic ratio analysis would help to estimate the contribution of gas burning, but this is out of the scope of the paper.

**R: Lines 494-496:** I don't find any relation between ERs and distance of sources in the manuscript. Is this a speculative result?

**A:** We observed that EF values under high ventilation conditions are lower than during low ventilation case (see Figure 6 and the related text). We explained this founding by the dilution of the CO2 peak that made the highest EF values (where the CO2 component is low) discarded from the distribution. Consequently, the ER distribution shifted towards the lower EFs during high ventilation cases. We could not attempt to find numerical relationship between the distance and the EF. We claimed that EF values during low ventilation cases are more relevant for the sources, so we used those values for the literature comparison.

---

## Author Response (AR2)

The authors thank for the anonymous referees their valuable comments and ideas. According to the recommendation of Referee #2, we have completed our manuscript with a section when the main error sources are discussed, and the final uncertainty of the method is estimated. Please see the section between Line 295 and 320 in the revised manuscript.